# Immune-Targeted Therapy with or without Transarterial Chemoembolization (TACE) for Advanced Hepatocellular Carcinoma with Portal Vein Tumor Thrombosis (PVTT): A Multicenter Retrospective Study

**DOI:** 10.3390/biomedicines12092124

**Published:** 2024-09-19

**Authors:** Ran You, Yuan Cheng, Lingfeng Diao, Chendong Wang, Bin Leng, Zeyu Yu, Qingyu Xu, Guowen Yin

**Affiliations:** 1Department of Interventional Oncology, Jiangsu Cancer Hospital, Jiangsu Institute of Cancer Research, Affiliated Cancer Hospital of Nanjing Medical University, Nanjing 210009, China; youran@njmu.edu.cn (R.Y.); lingfengdiao@163.com (L.D.); wangchendong@njmu.edu.cn (C.W.); lengbin9377@sina.com (B.L.); 02987b@njmu.edu.cn (Z.Y.); 2Department of Oncology, Bayi Hospital Affiliated to Nanjing University of Chinese Medicine, Nanjing 210009, China; chengyuan@csco.ac.cn

**Keywords:** hepatocellular carcinoma, portal vein tumor thrombus, transarterial chemoembolization, immune-targeted therapy, combination therapy

## Abstract

Purpose: In the present study, we aimed to assess the effectiveness and safety of immune-targeted therapy (IT) with or without transarterial chemoembolization (TACE) for advanced hepatocellular carcinoma (HCC) with portal vein tumor thrombosis (PVTT). Patients and methods: This was a multicenter retrospective study that included 265 HCC patients with PVTT (IT + TACE: 82, IT: 183). Overall survival (OS) and progression-free survival (PFS), as well as tumor responses and adverse events, were evaluated. Results: Patients in the IT + TACE group experienced significantly longer overall survival (OS) and progression-free survival (PFS) periods, compared with those in the IT group (OS 19.0 vs. 13.0 months, *p* < 0.001; PFS 12.0 vs. 7.3 months, *p* < 0.001). Multivariable analysis confirmed IT + TACE as an independent predictor for improved OS and PFS. Subgroup analysis demonstrated the benefits of IT + TACE in patients with rich PVTT blood supply. Preoperative imaging and DSA offered predictive value. Conclusions: TACE combined with IT provides a safe and effective treatment option for advanced-HCC patients with PVTT, particularly those with abundant PVTT blood supply.

## 1. Introduction

For Barcelona Clinic Liver Cancer (BCLC) stage C hepatocellular carcinoma (HCC), the standard treatment is systemic therapy [1]. However, regardless of whether the early-established sorafenib [2] or the currently favored lenvatinib [3], is used, the objective response rate (ORR) is less than 20%, and therapeutic efficacy is limited. In China, BCLC-C HCC is classified as a stage III disease, and guidelines recommend combining transarterial chemoembolization (TACE) with systemic therapy to achieve greater benefits [4]. The LAUNCH study showed the superiority of lenvatinib plus TACE over lenvatinib alone in terms of ORR, progression-free survival (PFS), and overall survival (OS) [5]. However, not all advanced HCC patients benefit from systemic therapy combined with TACE. In the TACTICS study [6], sorafenib plus TACE resulted in a greater PFS benefit than TACE alone, but no difference in OS was observed between groups, including the total population and the BCLC-C subgroup. Further investigation is needed to identify suitable candidates for systemic therapy combined with TACE for advanced HCC.

Portal vein tumor thrombosis (PVTT), as one of the characteristics of advanced liver cancer, has a significant impact on prognosis [7]. It is mainly supplied by the hepatic artery. After TACE, iodine oil can be deposited in the PVTT through the liver sinus, exerting a sustained antitumor effect. Studies have shown that patients with positive iodine oil deposition in PVTT after systemic therapy plus TACE experience longer OS periods than those with negative deposition, but the drawback is the need for post-TACE imaging re-evaluation, which lacks timeliness [8]. Identifying a suitable combination of systemic and local therapies also remains a challenge. The quality of PVTT blood supply directly affects the deposition of iodine oil in PVTT after TACE. The evaluation of PVTT blood supply mainly relies on preoperative imaging and intraoperative digital subtraction angiography (DSA), but there are currently no data on the blood supply of PVTT in patients treated with systemic therapy combined with TACE.

TACE in conjunction with targeted therapy represents a formidable strategy for the management of advanced HCC. Evidence suggests that this combined approach markedly improves OS when compared to either TACE or targeted therapy administered alone [9,10]. Our previous research has also demonstrated that the integration of TACE with systemic therapy further enhances the overall response rate (ORR) and survival outcomes in patients with unresectable HCC [11]. Nevertheless, the current evidence concerning the superiority of combination therapy over targeted therapy alone in HCC patients with PVTT remains scarce.

In the present study, we included BCLC-C HCC patients with PVTT to compare the efficacy of IT plus TACE versus IT alone, to identify the patient population that benefits most from IT plus TACE, and to evaluate the impact of PVTT blood supply in this population.

## 2. Materials and Methods

### 2.1. Study Design and Patients

The retrospective analysis was conducted on HCC patients with PVTT who were admitted to Jiangsu Cancer Hospital and the General Hospital of the Eastern Theater Command between March 2019 and September 2022, and who received IT with or without TACE. HCC diagnosis was based on clinical criteria established by the European Association for the Study of the Liver and the American Association for the Study of Liver Diseases. The main inclusion criteria were as follows: (1) patients aged 18–85 years; (2) first-line treatment with IT; (3) no tumor resection or liver transplantation after treatment; (4) Child–Pugh class A or B7; and (5) Eastern Cooperative Oncology Group performance status 0–1. The main exclusion criteria were as follows: (1) extrahepatic metastasis; (2) hepatic vein or inferior vena cava tumor thrombus; and (3) incomplete follow-up data. This retrospective study was approved by the Ethics Committee of Jiangsu Cancer Hospital (Approval No: 2022-076) and was conducted in accordance with the Helsinki Declaration. All patients provided informed consent for the use of their data and images.

### 2.2. PVTT Diagnosis and Classification

PVTT diagnosis was assessed by two radiologists with more than 10 years of experience, and was mainly based on the following imaging features [12]: (1) the presence of low signal mass in the portal vein; and (2) enhanced computed tomography (CT) or magnetic resonance imaging (MRI) indicating enhanced mass. PVTT was classified according to the VP classification from Japan [12], as follows: VP1, involving third-order branches or above; VP2, involving second-order branches; VP3, involving first-order branches; or VP4, involving the main portal vein or contralateral first-order branches.

### 2.3. IT

Targeted agents were administered at standard dosages, including tyrosine kinase inhibitors such as lenvatinib, sorafenib, and donafenib, as well as vascular endothelial growth factor (VEGF) inhibitors like bevacizumab. In instances where TACE is employed concomitantly, targeted therapy is initiated on the third day post-TACE, with dosage adjustments made in accordance with the product specifications and the patient’s body weight. The PD-1 inhibitors utilized include camrelizumab, sintilimab, pembrolizumab, and nivolumab, administered every three weeks. In the context of combination therapy with TACE, PD-1 inhibitors are commenced on the third day following the TACE procedure. In cases of severe treatment-related adverse events (TRAE), corticosteroids are employed. Should grade 3 or 4 TRAE persist, the PD-1 inhibitor is to be withheld. Upon alleviation of toxicity or when the patient exhibits tolerance to the treatment, re-administration may be considered.

### 2.4. TACE Procedure and PVTT Blood Supply Classification under DSA

The TACE protocol used in this study was previously described. TACE was performed by interventional radiologists, each with over 5 years of experience. After local anesthesia, a 5F catheter (COOK) was used for hepatic artery angiography and indirect portal vein angiography to assess tumor blood supply and portal vein conditions. Rich-blood-type PVTT was defined as follows: During arterial phase angiography, PVTT showed abundant blood supply, with tumor vessels originating from hepatic artery branches, exhibiting a striped pattern; during the venous phase, the tumor thrombus showed obvious tumor staining. Poor-blood-type PVTT was defined as follows: During arterial phase angiography, the tumor thrombus showed sparse, small, irregular striped patterns or no blood supply arteries; during the parenchymal phase, the tumor thrombus showed unclear or no tumor staining. After excluding severe arteriovenous shunts, a 2.7F microcatheter (Terumo, Tokyo, Japan) was used to super-selectively embolize the tumor artery. Depending on tumor size and distribution, the following protocol was used: a mixture of 20 mg epirubicin and 10 mL iodized oil, with a volume ratio of 2:1. The amount of iodized oil injected was determined by tumor size and vascular conditions. After embolization, the tumor blood supply was reduced, and embolic agents (gelatin sponge particles, microspheres, and polyvinyl alcohol) were then used to block the tumor-feeding artery. The embolization endpoint was the cessation of blood flow in the tumor-feeding artery. This was followed by angiography to confirm the distribution of iodized oil and exclude ectopic embolism.

### 2.5. Follow-Up and Outcomes

All patients received routine follow-up after treatment, including enhanced CT/MRI and laboratory tests (blood routine, liver and kidney function, and alpha-fetoprotein (AFP) level). Evaluation was conducted by two radiologists, each with over 10 years of experience. According to the modified Response Evaluation Criteria in Solid Tumors, tumor response was categorized as complete response (CR), partial response (PR), stable disease (SD), or progressive disease (PD). Follow-up was conducted in September 2023.

Outcomes included OS, PFS, ORR, disease control rate (DCR), and adverse events (AEs). ORR was defined as the proportion of patients with CR or PR. DCR was defined as the proportion of patients with CR, PR, or SD. OS was defined as the time from the initiation of treatment to the date of death. PFS was defined as the time from the initiation of the treatment to disease progression or death.

### 2.6. Statistical Analysis

Continuous variables were expressed as median (interquartile range (IQR)), and comparison between groups was conducted using the Mann–Whitney U test. For categorical variables, chi-square testing or Kruskal–Wallis testing was used for comparison between groups. The Kaplan–Meier method and log-rank testing were used to compare OS and PFS. Univariable Cox regression analysis was used to screen potential associated factors for OS and PFS. Variables with *p* < 0.1 or previously considered predictors of survival were included in the multivariable Cox regression analysis. Receiver operating characteristic (ROC) curves and area under the curve (AUC) values were used to measure predictive performance. Statistical analysis was performed using SPSS 25.0 (IBM Corp, Chicago, IL, USA). *p* < 0.05 was considered statistically significant.

## 3. Results

### 3.1. Patient Characteristics

A total of 354 HCC patients with BCLC stage C and PVTT who received IT + TACE or IT were screened. Among them, 89 patients were excluded due to meeting the exclusion criteria (Figure 1). Finally, 265 patients were included in this study (82 in the IT + TACE group, and 183 in the IT group). There were no significant differences between the two groups in terms of baseline characteristics, including age, sex, HBV, Child–Pugh class, albumin–bilirubin (ALBI), cirrhosis, AFP level, number of tumor lesions, tumor size, and PVTT classification (Table 1).

### 3.2. Treatment Responses

In the IT + TACE and IT groups, respectively, the CR rates were 2.4% and 0%, the PR rates were 61.0% and 26.8%, and ORR values were 63.4% and 26.8% (Table 2). Tumor response in the IT + TACE group was significantly better than that in the IT group (*p* < 0.001).

Patients in the IT + TACE group experienced significantly longer survival periods, compared with those in the IT group (Figure 2). Median OS periods for the IT-TACE group and IT group were 19.0 months and 13.0 months (hazard ratio (HR): 0.34, 95% confidence interval (CI): 0.24–0.49, *p* < 0.001), respectively; median PFS periods were 12.0 months and 7.3 months (HR: 0.28, 95% CI: 0.19–0.41, *p* < 0.001), respectively.

### 3.3. Associated Factors for OS and PFS

The multivariable Cox proportional hazards model showed that IT combined with TACE was independently associated with longer OS periods (HR: 0.31, 95% CI: 0.21–0.46, *p* < 0.001) and PFS periods (HR: 0.28, 95% CI: 0.19–0.42, *p* < 0.001) (Table 3 and Table 4). Additionally, Child–Pugh class, tumor size, PVTT classification, and MRI arterial enhancement were independently associated factors for OS. Independently associated factors for PFS included Child–Pugh class, cirrhosis, PVTT classification, and MRI arterial enhancement.

Subgroup analysis in the forest plot indicated that, in most subgroups, the median OS and PFS in the IT-TACE group were superior to those in the IT group (Figure 3). Further analysis of PVTT imaging enhancement showed that rich-blood-supply PVTT achieved better efficacy in the combination group, but no significant difference was exhibited in the monotherapy group (Table 5).

Beyond the therapeutic modalities, the efficacy of treatment in relation to the grading of PVTT merits further investigation. We compared OS and PFS among patients classified as VP2, VP3, and VP4. Our findings indicated a significant correlation between VP grading and patient prognosis, regardless of the treatment modality employed (Appendix A). Notably, higher VP grades were associated with poorer prognoses. Furthermore, we examined the OS and PFS related to portal vein thrombus grading in both the combination therapy group and the sole targeted immunotherapy group. The data reveal that OS in the combination group correlated with VP grading, whereas PFS exhibited no significant differences (Appendix A). Conversely, in the targeted immunotherapy group, PFS was significantly associated with VP grading, while OS displayed no such differences (Appendix A).

Our study showed that the enhancement patterns of PVTT on MRI and DSA were positively correlated (r = 0.43, *p* < 0.001). ROC analysis indicated the potential diagnostic value of the enhancement patterns of PVTT on MRI and DSA for distinguishing the post-treatment efficacy of TACE combined with IT for HCC. The AUC for the enhancement pattern of PVTT was 0.7026 (95%CI: 0.6103–0.7948) on MRI, and 0.7769 (95%CI: 0.6918–0.8620) on DSA, with similar predictive performance (*p* = 0.1503). After combining MRI and DSA, the AUC for the enhancement patterns of PVTT was 0.8109 (95%CI: 0.7229–0.8989) (Figure 4).

### 3.4. Safety

Table 6 lists all treatment-related AEs (TRAEs) in the two groups, including diarrhea, hand–foot syndrome, hypertension, fatigue, anorexia and nausea, rash, oral ulcer, trachyphonia, thyroid dysfunction, hyperbilirubinemia, proteinuria, and decreased platelet count. There were no differences in grade 3 or 4 TRAEs between the two groups. No grade 5 TRAEs were observed in either group.

## 4. Discussion

In this multicenter retrospective cohort study, we focused on patients with advanced HCC and PVTT. Our results showed that TACE combined with IT resulted in longer OS and PFS periods, as well as higher ORR values, compared with IT alone. Furthermore, the multivariable analysis indicated that TACE combined with IT was an independent associated factor for prolonging OS and PFS.

The study showed that TACE combined with IT was more effective than IT alone (median OS: 19.0 vs. 13.0 months; median PFS: 12.0 vs. 7.3 months; ORR: 63.4% vs. 26.8%). These data support the conclusion of both the LAUNCH and CHANCE2211 studies that TACE combined with systemic therapy is superior to systemic therapy alone [5,10]. The median OS in the IT + TACE group was 19.0 months; this is numerically higher than the 17.8 months reported for lenvatinib plus TACE in the LAUNCH study. The reason for this might be that our enrolled patients received IT, whereas the patients in the LAUNCH study received only targeted therapy. The median OS with IT + TACE in the present study was similar to that achieved with TACE plus IT in CHANCE2211 (24.1 months). However, the median OS in our IT group was 13.0 months, a lower figure than the 19.2 months in the IMbrave150 data [13]. The main explanation for this might be that all the patients in our study had BCLC-C disease, whereas 15% of the patients in Imbrave150 had BCLC-B disease, and this subgroup of patients experienced longer OS periods. In our study, the ORR of IT + TACE was 63.4%; this was comparable to previous results [11,14], and higher than the 26.8% we obtained for IT alone. The explanation for this finding lies in the fact that TACE treatment alters the immune microenvironment of the tumor, turning immunosuppressive cold tumors into hot tumors and enhancing immunity [15]. At the same time, TACE blocks the tumor blood supply to promote the expression of angiogenic factors such as VEGF, which has a synergistic effect when combined with anti-vascular targeted drugs [16].

Recent investigations into local and targeted HCC accompanied by PVTT have emerged as a focal point in contemporary research. Yuan [17] recently examined the efficacy of TACE-HAIC combined with targeted immunotherapy compared to conventional TACE treatment in patients with HCC and PVTT. While the overarching approach bears similarities, our study diverges in the control populations utilized. Theirs employed a control group receiving only TACE, whereas our control cohort comprises patients undergoing solely targeted immunotherapy. Both control groups are reflective of the prevalent clinical protocols for HCC treatment today. Despite the differences in patient populations, our findings resonate consistently with theirs; the combination of TACE and targeted immunotherapy offers greater benefits compared to monotherapy options—whether TACE alone or targeted immunotherapy alone—thus widening the eligibility criteria for treatment and rendering our conclusions more clinically relevant and broadly applicable. In addition, another piece of literature evaluates the comparative efficacy of local combined targeted immunotherapy regimens in patients with HCC and PVTT [18]. While the targeted immunotherapy protocols are consistent, the primary distinction lies in the local treatment modality being either TACE or HAIC. Their conclusions indicate that HAIC offers superior survival and therapeutic advantages as a combined treatment regimen, a population notably distinct from our exclusively TACE cohort. This suggests a promising avenue for future clinical research, whereby the selection of HAIC could yield enhanced clinical benefits for this patient demographic and provide further empirical support for the paradigm of localized combined systemic treatment approaches.

Previous studies have explored the efficacy and safety of localized therapies and combination regimens in the treatment of advanced HCC [5,9]. However, our research delineates several distinguishing characteristics. Prior investigations often encompassed a heterogeneous population of advanced HCC patients, including those with portal vein invasion, lymph node metastasis, and various distant metastases. In contrast, our study exclusively focuses on cases of HCC associated solely with PVTT, systematically excluding patients with distant metastases or lymph node involvement, thereby targeting a more homogenous group. Moreover, we have conducted a profound subgroup analysis regarding the vascular supply of PVTT. This represents the inaugural statistical examination of targeted therapies or combination treatments predicated on the vascular characteristics of PVTT, enabling a stratified approach within the PVTT patient cohort. Our findings aim to identify subpopulations that may benefit most, thereby providing more precise clinical guidance.

Previous studies have explored the predictive value of iodized oil deposition within PVTT for the efficacy of TACE in treating advanced HCC [8]. However, these studies were limited by poor timeliness. In the present study, we attempted to enhance timeliness by predicting prognosis through pre-treatment imaging assessment of PVTT blood supply. Additionally, in the present study, we used pre-treatment MRI and DSA data to assess PVTT blood supply and predict prognosis.

The main blood supply of PVTT comes from the hepatic artery. After TACE treatment, the iodine oil occludes the tumor blood flow, leading to necrosis of the PVTT, and thereby improving the prognosis [19]. This was confirmed in the present study, which showed that the richer the blood supply to PVTT, the better the effect of TACE treatment. This may have been because the richer the blood supply to the tumor thrombus, the greater the deposits of iodine oil inside PVTT after TACE. This causes the tumor thrombus to necrotize, and results in a better prognosis. In the present study, we also found that the blood supply to PVTT did not affect the efficacy of IT alone. This may have been because IT controls tumor growth by inhibiting tumor blood vessels, but cannot achieve complete ischemic necrosis of the tumor by occluding tumor blood vessels, as TACE does. We may say, therefore, that the quality of PVTT blood vessels had little impact on prognosis in the IT group.

Our data showed that TACE combined with IT can deliver better survival outcomes. However, there is currently no consensus on which type of patients can most benefit from this combination therapy. Forest plot analysis of OS and PFS indicated that the IT + TACE group gained survival benefits across almost all subgroups, compared with IT alone, especially in subgroups categorized by AFP level, PVTT classification, cirrhosis, and MRI arterial enhancement, in line with previous studies. It is worth noting that patients with VP3 and VP4 subtypes of PVTT in the IT + TACE group gained significant benefits, but this trend was not observed in VP2 patients. This may have been due to the small number of VP2 patients and large fluctuations in HR, as well as certain biases. Further investigation is needed, involving greater numbers of such patients, if more precise conclusions are to be obtained.

In the present study, the TRAEs in both groups were controllable, and no treatment-related deaths occurred. The most common TRAEs were hypertension, hand–foot syndrome, and fatigue. Frequencies and types were similar to those reported in previous studies. In addition, following the principles of on-demand treatment and precision TACE, liver function was actively protected. The incidence of grade 3–4 TRAEs in the IT + TACE group was slightly higher than that in the IT group, but no statistical difference was recorded, indicating that the introduction of TACE does not significantly increase the risk of AEs. Our findings lend support to the notion that TACE does not exacerbate the risk of hepatic failure in patients with HCC accompanied by PVTT. Anatomically, the liver receives a predominant portion of its blood supply via the portal vein. The presence of portal vein thrombosis poses a risk for hepatic ischemia and subsequent liver failure following TACE, which entails embolization of the hepatic artery. According to Chinese clinical guidelines [20], portal vein tumor thrombosis is not an absolute contraindication for TACE. However, in cases where the main trunk thrombus leads to severe hepatic dysfunction, TACE should be avoided. The patients included in our study exhibited satisfactory hepatic reserve. Despite imaging evidence of thrombus, collateral circulation had compensated over an extended period, rendering their liver function suitable for the consideration of TACE.

This article represents a retrospective study in which the systemic treatment protocols for patients were not uniformly established. The selection of systemic treatment regimens merits further deliberation, particularly given the multitude of therapeutic options for advanced HCC. The chronological advancement from the initial monotherapy of sorafenib, through lenvatinib, to various combination therapies underscores the complexity of first-line treatment selections, a matter that is continuously debated in terms of optimizing patient outcomes. Lenvatinib, as a multi-targeted inhibitor, demonstrates the highest ORR among single-target kinase inhibitors; however, the translation of this elevated ORR into prolonged OS benefits remains inadequately explored. A meta-analysis [21] indicates that, while lenvatinib may yield superior ORR and PFS, these advantages do not necessarily confer survival benefits. Possible explanations for this discrepancy may include the proportion of patients receiving subsequent second-line therapies, as well as the impact of hepatic function impairment and other confounding factors influencing OS outcomes.

This study was a retrospective study that was affected by selection bias and a limited sample size. In addition, the types of targeted therapy drugs in the study population were not uniform, and our conclusions need to be confirmed by better prospective randomized studies. Moreover, patients with extrahepatic metastases, hepatic vein, and inferior vena cava tumor thrombus were excluded from the present study. Our study population mainly included patients with hepatitis B. The question of whether our conclusions are applicable to HCC patients with other etiologies such as hepatitis C and alcoholic liver disease can only be answered after further investigation.

## 5. Conclusions

In advanced HCC patients with PVTT, TACE combined with IT was found to be more effective than IT alone. The combination treatment was associated with higher ORRs, and longer periods of PFS and OS. In addition, its safety profile was manageable. Patients with abundant blood supply to PVTT can benefit from TACE combined with IT, and preoperative imaging combined with DSA can be used to better assess the blood supply to PVTT, and more effectively predict prognosis.

## Figures and Tables

**Figure 1 biomedicines-12-02124-f001:**
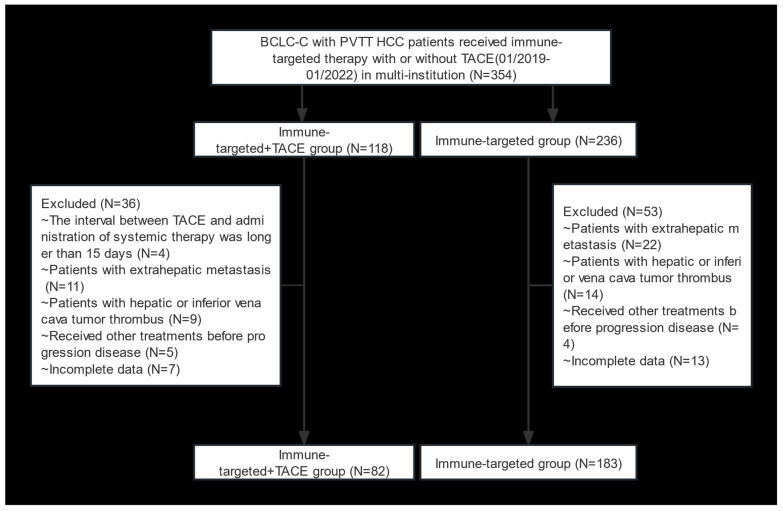
Flow diagram of selection criteria. Abbreviations: BCLC, Barcelona Clinic Liver Cancer; PVTT, portal vein tumor thrombus; HCC, hepatocellular carcinoma; TACE, transarterial chemoembolization.

**Figure 2 biomedicines-12-02124-f002:**
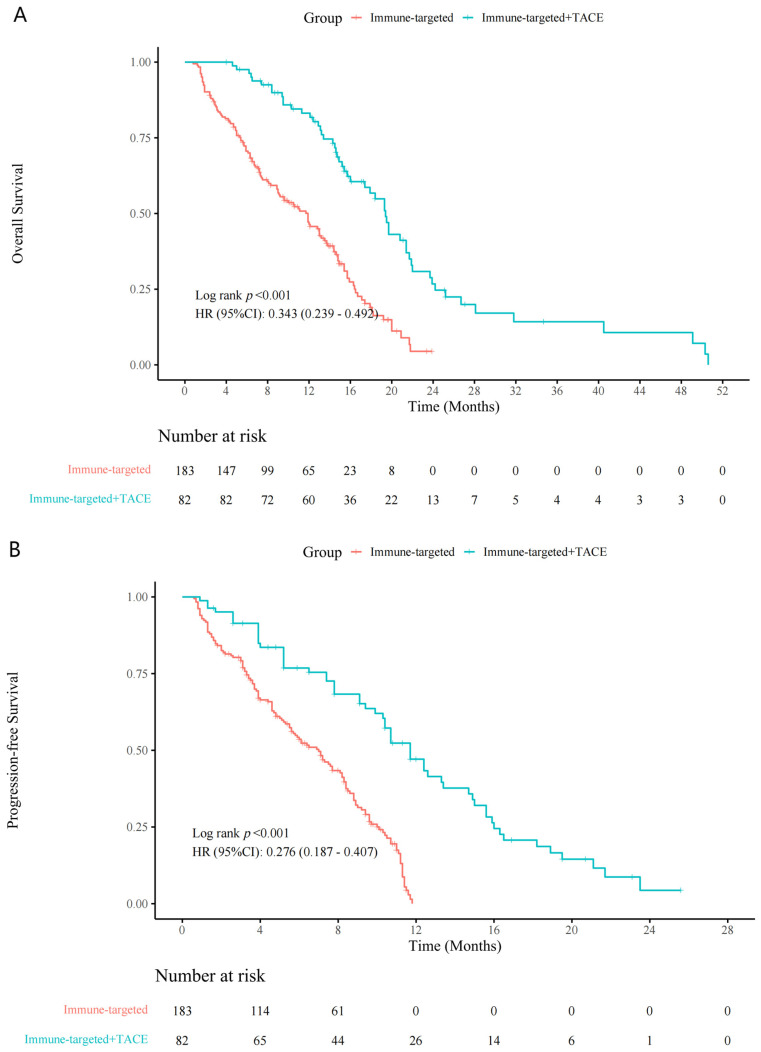
Kaplan–Meier curves for (**A**) overall survival and (**B**) progression-free survival in the IT + TACE group and the IT group. Abbreviations: IT—immune-targeted therapy; TACE—transarterial chemoembolization.

**Figure 3 biomedicines-12-02124-f003:**
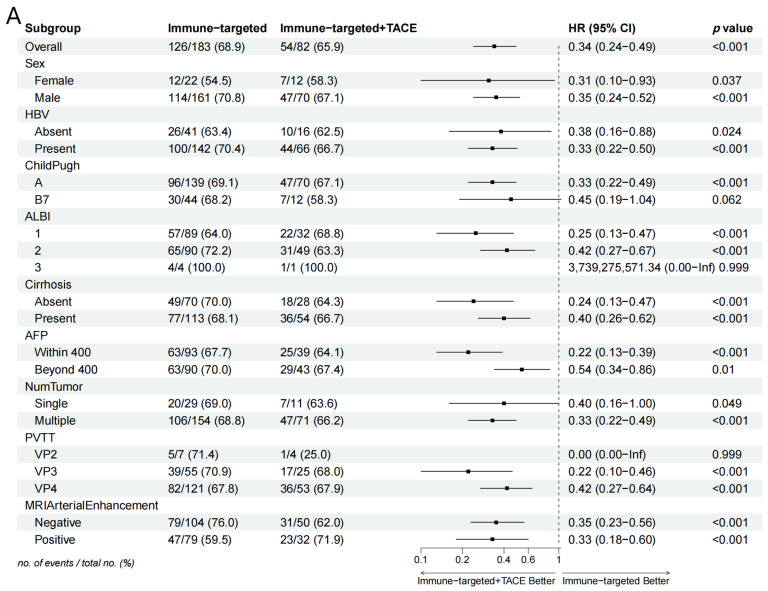
Forest plots of (**A**) overall survival and (**B**) progression-free survival in subgroups of patients treated with IT + TACE and IT. Abbreviations: TACE—transarterial chemoembolization; HR—hazard ratio; CI—confidence interval; HBV—hepatitis B virus; ALBI—albumin–bilirubin; AFP—alpha-fetoprotein; PVTT—portal vein tumor thrombus.

**Figure 4 biomedicines-12-02124-f004:**
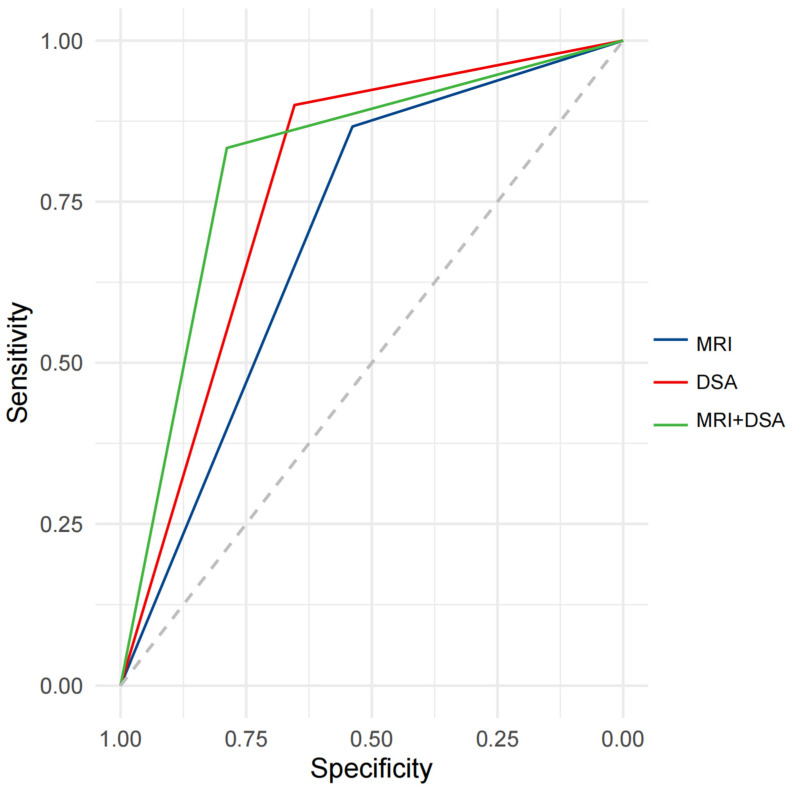
ROC curve analysis for predicting efficacy after treatment with IT + TACE. Abbreviations: ROC—receiver operating curve; MRI—magnetic resonance imaging; DSA—digital subtraction angiography.

**Table 1 biomedicines-12-02124-t001:** Baseline characteristics.

Characteristics	IT + TACE (*n* = 82)	IT (*n* = 183)	*p* Value
**Age (years), median (IQR)**	59 (48, 66)	55 (51, 64)	0.509
**Sex**			0.557
Male	70 (85.4%)	161 (88.0%)	
Female	12 (14.6%)	22 (12.0%)	
**HBV**			0.596
Absent	16 (19.5%)	41 (22.4%)	
Present	66 (80.5%)	142 (77.6%)	
**Child–Pugh class**			0.083
A	70 (85.4%)	139 (76.0%)	
B7	12 (14.6%)	44 (24.0%)	
**ALBI**			0.263
1	32 (39.0%)	89 (48.6%)	
2	49 (59.8%)	90 (49.2%)	
3	1 (1.2%)	4 (2.2%)	
**Cirrhosis**			0.522
Absent	28 (34.1%)	70 (38.3%)	
Present	54 (65.9%)	113 (61.7%)	
**AFP level (ng/mL** **)**			0.624
Within 400	39 (47.6%)	93 (50.8%)	
Beyond 400	43 (52.4%)	90 (49.2%)	
**Number of tumor lesions**			0.609
1	11 (13.4%)	29 (15.8%)	
≥2	71 (86.6%)	154 (84.2%)	
**Tumor size (cm), median (IQR)**	7.3 (4.0, 11.9)	8.6 (5.1, 11.6)	0.185
**PVTT classification**			0.886
VP2	4 (4.9%)	7 (3.8%)	
VP3	25 (30.5%)	55 (30.1%)	
VP4	53 (64.6%)	121 (66.1%)	

Note: Results are presented as *n* (%), unless otherwise indicated. Abbreviations: IT—immune-targeted therapy; TACE—transarterial chemoembolization; IQR—interquartile range; HBV—hepatitis B virus; ALBI—albumin–bilirubin; AFP—alpha-fetoprotein; PVTT—portal vein tumor thrombus.

**Table 2 biomedicines-12-02124-t002:** Tumor responses.

Response	IT + TACE (*n* = 82)	IT (*n* = 183)	*p* Value
**Best response**			<0.001
CR	2 (2.4%)	0	
PR	50 (61.0%)	49 (26.8%)	
SD	25 (30.5%)	77 (42.1%)	
PD	5 (6.1%)	57 (31.1%)	
**ORR**	52 (63.4%)	49 (26.8%)	<0.001
**DCR**	77 (93.9%)	126 (68.9%)	<0.001

Note: Results are presented as *n* (%). Abbreviations: CR—complete response; DCR—disease control rate; IT—immune-targeted therapy; ORR—objective response rate; PD—progressive disease; PR—partial response; SD—stable disease; TACE—transarterial chemoembolization.

**Table 3 biomedicines-12-02124-t003:** Cox regression analyses of associated factors for overall survival.

Characteristics	Univariable Analysis					Multivariable Analysis				
	N	Events	HR	95% CI	*p* Value	N	Events	HR	95% CI	*p* Value
**Treatment group**										
IT + TACE	82	54	—	—		82	54	—	—	
IT	183	126	2.91	2.03, 4.18	<0.001	183	126	3.14	2.17, 4.55	<0.001
**Age**	265	180	1.00	0.98, 1.01	0.705					
**Sex**										
Male	231	161	—	—						
Female	34	19	0.78	0.49, 1.26	0.312					
**HBV**										
Absent	57	36	—	—						
Present	208	144	1.11	0.77, 1.60	0.589					
**Child–Pugh class**										
A	209	143	—	—		209	143	—	—	
B7	56	37	2.14	1.48, 3.09	<0.001	56	37	1.94	1.33, 2.82	<0.001
**ALBI**										
1	121	79	—	—						
2	139	96	1.07	0.79, 1.44	0.663					
3	5	5	2.13	0.86, 5.28	0.103					
**Cirrhosis**										
Absent	98	67	—	—		98	67	—	—	
Present	167	113	1.39	1.02, 1.89	0.036	167	113	1.32	0.96, 1.81	0.085
**AFP level**										
Within 400	132	88	—	—						
Beyond 400	133	92	1.31	0.97, 1.77	0.077					
**Tumor size**	265	180	1.05	1.01, 1.08	0.005	265	180	1.03	1.00, 1.07	0.044
**Number of tumor lesions**										
1	40	27	—	—		40	27	—	—	
≥2	225	153	1.26	0.84, 1.91	0.264	225	153	1.38	0.90, 2.11	0.139
**PVTT classification**										
VP2	11	6	—	—		11	6	—	—	
VP3	80	56	2.36	1.01, 5.51	0.047	80	56	2.70	1.14, 6.40	0.024
VP4	174	118	3.04	1.33, 6.95	0.008	174	118	3.21	1.39, 7.43	0.006
**MRI arterial enhancement**										
Negative	154	110	—	—		154	110	—	—	
Positive	111	70	0.74	0.55, 1.00	0.054	111	70	0.67	0.50, 0.91	0.011

Abbreviations: HR—hazard ratio; CI—confidence interval; TACE—transarterial chemoembolization; HBV—hepatitis B virus; ALBI—albumin–bilirubin; AFP—alpha-fetoprotein; PVTT—portal vein tumor thrombus.

**Table 4 biomedicines-12-02124-t004:** Cox regression analyses of associated factors for progression-free survival.

Characteristics	Univariable Analysis					Multivariable Analysis				
	N	Events	HR	95% CI	*p* Value	N	Events	HR	95% CI	*p* Value
**Treatment group**										
IT + TACE	82	57	—	—		82	57	—	—	
IT	183	143	3.62	2.46, 5.34	<0.001	183	143	3.77	2.53, 5.60	<0.001
**Age**	265	200	1.00	0.98, 1.01	0.671					
**Sex**										
Male	231	176	—	—		231	176	—	—	
Female	34	24	0.80	0.52, 1.23	0.319	34	24	0.71	0.46, 1.09	0.115
**HBV**										
Absent	57	40	—	—						
Present	208	160	1.05	0.75, 1.49	0.763					
**Child–Pugh class**										
A	209	154	—	—		209	154	—	—	
B7	56	46	2.82	2.00, 3.98	<0.001	56	46	2.64	1.85, 3.76	<0.001
**ALBI**										
1	121	93	—	—						
2	139	103	1.10	0.83, 1.46	0.520					
3	5	4	2.84	1.03, 7.80	0.043					
**Cirrhosis**										
Absent	98	78	—	—		98	78	—	—	
Present	167	122	1.39	1.04, 1.86	0.024	167	122	1.36	1.01, 1.83	0.040
**AFP level**										
Within 400	132	102	—	—						
Beyond 400	133	98	1.25	0.94, 1.65	0.127					
**Tumor size**	265	200	1.03	1.00, 1.06	0.031					
**Number of tumor lesions**										
1	40	30	—	—		40	30	—	—	
≥2	225	170	1.31	0.89, 1.94	0.171	225	170	1.44	0.97, 2.15	0.071
**PVTT classification**										
VP2	11	8	—	—		11	8	—	—	
VP3	80	65	2.42	1.16, 5.08	0.019	80	65	2.38	1.13, 5.04	0.023
VP4	174	127	3.03	1.47, 6.26	0.003	174	127	3.00	1.44, 6.26	0.003
**MRI arterial enhancement**										
Negative	154	114	—	—		154	114	—	—	
Positive	111	86	0.80	0.60, 1.06	0.119	111	86	0.74	0.56, 0.99	0.041

Abbreviations: HR—hazard ratio; CI—confidence interval; TACE—transarterial chemoembolization; HBV—hepatitis B virus; ALBI—albumin–bilirubin; AFP—alpha-fetoprotein; PVTT—portal vein tumor thrombus.

**Table 5 biomedicines-12-02124-t005:** Comparison of baseline characteristics and tumor responses between patients with and without MRI arterial enhancement in each group.

Characteristic	IT + TACE			IT		
	MRI Arterial Enhancement			MRI Arterial Enhancement		
	Negative (*n* = 50)	Positive (*n* = 32)	*p* Value	Negative (*n* = 104)	Positive (*n* = 79)	*p* Value
**Age (years), median (IQR)**	58 (47, 67)	59 (52, 66)	0.379	55 (51, 63)	54 (51, 64)	0.516
**Sex**			0.757			0.492
Male	42 (84.0%)	28 (87.5%)		90 (86.5%)	71 (89.9%)	
Female	8 (16.0%)	4 (12.5%)		14 (13.5%)	8 (10.1%)	
**HBV**			0.477			0.334
Absent	11 (22.0%)	5 (15.6%)		26 (25.0%)	15 (19.0%)	
Present	39 (78.0%)	27 (84.4%)		78 (75.0%)	64 (81.0%)	
**Child–Pugh class**			0.757			0.728
A	42 (84.0%)	28 (87.5%)		78 (75.0%)	61 (77.2%)	
B7	8 (16.0%)	4 (12.5%)		26 (25.0%)	18 (22.8%)	
**ALBI**			0.606			0.337
1	20 (40.0%)	12 (37.5%)		46 (44.2%)	43 (54.4%)	
2	30 (60.0%)	19 (59.4%)		56 (53.8%)	34 (43.0%)	
3	0	1 (3.1%)		2 (1.9%)	2 (2.5%)	
**Cirrhosis**			0.322			0.946
Absent	15 (30.0%)	13 (40.6%)		40 (38.5%)	30 (38.0%)	
Present	35 (70.0%)	19 (59.4%)		64 (61.5%)	49 (62.0%)	
**AFP level (ng/mL)**			0.723			0.732
Within 400	23 (46.0%)	16 (50.0%)		54 (51.9%)	39 (49.4%)	
Beyond 400	27 (54.0%)	16 (50.0%)		50 (48.1%)	40 (50.6%)	
**Number of tumor lesions**			>0.999			0.311
1	7 (14.0%)	4 (12.5%)		14 (13.5%)	15 (19.0%)	
≥2	43 (86.0%)	28 (87.5%)		90 (86.5%)	64 (81.0%)	
**PVTT classification**			>0.999			0.626
VP2	3 (6.0%)	1 (3.1%)		3 (2.9%)	4 (5.1%)	
VP3	15 (30.0%)	10 (31.3%)		30 (28.8%)	25 (31.6%)	
VP4	32 (64.0%)	21 (65.6%)		71 (68.3%)	50 (63.3%)	
**Tumor size (cm), median (IQR)**	7.2 (3.6, 11.2)	7.7 (4.9, 12.6)	0.276	8.5 (5.1, 11.2)	8.7 (5.4, 11.7)	0.723
**Best response**			<0.001			0.500
CR	1 (2.0%)	1 (3.1%)		0	0	
PR	23 (46.0%)	27 (84.4%)		26 (25.0%)	23 (29.1%)	
SD	22 (44.0%)	3 (9.4%)		42 (40.4%)	35 (44.3%)	
PD	4 (8.0%)	1 (3.1%)		36 (34.6%)	21 (26.6%)	

Note: Results are presented as *n* (%), unless otherwise indicated. Abbreviations: CR—complete response; IT—immune-targeted therapy; PD—progressive disease; PR—partial response; SD—stable disease; TACE—transarterial chemoembolization; IQR—interquartile range; HBV—hepatitis B virus; ALBI—albumin–bilirubin; AFP—alpha-fetoprotein; PVTT—portal vein tumor thrombus.

**Table 6 biomedicines-12-02124-t006:** Treatment-related adverse events.

Events	IT + TACE (*n* = 82)		IT (*n* = 183)		*p* Value
	Any Grade	Grade 3–4	Any Grade	Grade 3–4	
Diarrhea	10 (12.2%)	3 (3.7%)	27 (14.8%)	10 (5.5%)	>0.999
Hand–foot syndrome	20 (24.4%)	9 (11.0%)	30 (16.4%)	9 (4.9%)	0.462
Hypertension	15 (18.3%)	5 (6.1%)	39 (21.3%)	7 (3.8%)	0.489
Fatigue	23 (28.0%)	4 (4.9%)	58 (31.7%)	20 (10.9%)	0.248
Anorexia and nausea	23 (28.0%)	5 (6.1%)	37 (20.2%)	7 (3.8%)	>0.999
Rash	18 (22.0%)	3 (3.7%)	34 (18.6%)	6 (3.3%)	>0.999
Oral ulcer	13 (15.9%)	6 (7.3%)	45 (24.6%)	12 (6.6%)	0.364
Trachyphonia	6 (7.3%)	0	24 (13.1%)	9 (4.9%)	0.305
Thyroid dysfunction	8 (9.8%)	5 (6.1%)	18 (9.8%)	3 (1.6%)	0.211
Hyperbilirubinemia	26 (31.7%)	11 (13.4%)	50 (27.3%)	16 (8.7%)	0.544
Proteinuria	10 (12.2%)	3 (3.7%)	11 (6.0%)	2 (1.1%)	>0.999
Platelet count decreased	16 (19.5%)	4 (4.9%)	22 (12.0%)	14 (7.7%)	0.147

Note: Results are presented as *n* (%). Abbreviations: IT—immune-targeted therapy; TACE—transarterial chemoembolization.

## Data Availability

The data presented in this study are available upon request from the corresponding author (Guowen Yin). They are not publicly available, to maintain patient privacy.

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
