# Peer review of "Immune-Targeted Therapy with or without Transarterial Chemoembolization (TACE) for Advanced Hepatocellular Carcinoma with Portal Vein Tumor Thrombosis (PVTT): A Multicenter Retrospective Study"

_biomedicines, 2024, doi:10.3390/biomedicines12092124_

Round 1

Reviewer 1 Report

Comments and Suggestions for Authors

The Authors present the results of a retrospective study investigating the combined use of targeted therapy and TACE in patients with HCC and PVTT.

the study is overall well conducted, although similar studies have already been published and should be cited (Yuan Y et al, Int J Surg. 2023; Lin Z et al, Am J Cancer Res 2023).

My main comment is, can the Authors better clarify the medical treatment used and stratify data accordingly?

some minor comments:

- English needs revision

- quality of figures must be improved

- Lines 215-217, I guess they are an oversight

Comments on the Quality of English Language

Extensive revision needed

Author Response

Comments 1:the study is overall well conducted, although similar studies have already been published and should be cited (Yuan Y et al, Int J Surg. 2023; Lin Z et al, Am J Cancer Res 2023).
Response 1:I am grateful to the reviewers for their insightful guidance and feedback. They meticulously reviewed numerous references and highlighted two studies relevant to our work. The first, by Yuan Y et al. (Int J Surg. 2023, PMID: 37026861), investigates the comparative efficacy of TACE-HAIC combined with targeted immunotherapy versus TACE alone in patients with hepatic carcinoma and portal vein tumor thrombus. Although their study's overall approach is akin to ours, their control group received only TACE, while ours received only targeted immunotherapy. Both control groups represent current mainstream clinical protocols for hepatic carcinoma. Despite the differences in study populations, the conclusions align: combining TACE with targeted immunotherapy offers greater benefits compared to single treatment modalities, enhancing its clinical relevance and applicability. This discussion has been incorporated and cited in the manuscript (line 284-296,References 17).
The second study referenced by the reviewers (PMID: 38058801) compares local combined therapies with targeted immunotherapy in patients with hepatic carcinoma and portal vein tumor thrombus, differing primarily in the local treatment modality—TACE versus HAIC. Their findings suggest that HAIC as part of a combined treatment offers superior survival and efficacy benefits. This contrasts with our study, which exclusively involved TACE patients. This provides a novel perspective and suggests that incorporating HAIC could yield greater clinical benefits for such patients. Future clinical research might explore this possibility, contributing further evidence to the concept of combining local and systemic therapies. This analysis has also been added to the discussion and cited (line 296-305,References 18).

Comments 2:My main comment is, can the Authors better clarify the medical treatment used and stratify data accordingly?
Response 2:In the methods section of our manuscript, we delineated the approaches for targeted therapy and TACE; however, the discourse on the treatment duration of systemic therapeutics, the adjustment protocols following adverse reactions, and the synergistic application of systemic agents with TACE remained insufficiently articulated. We have therefore incorporated additional details on these aspects.
Targeted agents were administered at standard dosages, including tyrosine kinase inhibitors such as Lenvatinib, Sorafenib, and Donafenib, as well as vascular endothelial growth factor (VEGF) inhibitors like Bevacizumab. In instances where TACE is employed concomitantly, targeted therapy is initiated on the third day post-TACE, with dosage adjustments made in accordance with the product specifications and the patient's body weight. The PD-1 inhibitors utilized include Camrelizumab, Sintilimab, Pembrolizumab, and Nivolumab, administered every three weeks. In the context of combination therapy with TACE, PD-1 inhibitors are commenced on the third day following the TACE procedure. In cases of severe treatment-related adverse events (TRAE), corticosteroids are employed. Should grade 3 or 4 TRAE persist, the PD-1 inhibitor is to be withheld. Upon alleviation of toxicity or when the patient exhibits tolerance to the treatment, re-administration may be considered (line 94-105).
Regarding data stratification, we appreciate the valuable suggestion. Recognizing the limited data for subgroup analysis, we have augmented the results section with an analysis of the various subgroups of portal vein tumor thrombus (PVTT). The relevant data and figures are presented in the results section and supplementary figures, as follows:
Beyond the therapeutic modalities, the efficacy of treatment in relation to the grading of PVTT merits further investigation. We compared OS and PFS among patients classified as VP2, VP3, and VP4. Our findings indicate a significant correlation between VP grading and patient prognosis, regardless of the treatment modality employed (Supplementary Figures 1A and 1B). Notably, higher VP grades were associated with poorer prognoses. Furthermore, we examined the OS and PFS related to portal vein thrombus grading in both the combination therapy group and the sole targeted immunotherapy group. The data reveal that OS in the combination group correlated with VP grading, whereas PFS exhibited no significant differences (Supplementary Figures 2A and 2B). Conversely, in the targeted immunotherapy group, PFS was significantly associated with VP grading, while OS displayed no such differences (Supplementary Figures 3A and 3B)(line 201-212).

Comments 3:English needs revision
Response 3:I am from a non-English-speaking country, and my English expressions may not be idiomatic. I have sought revision services from MDPI (english-84473) to enhance the clarity of my writing. Further refinement to elevate the expression to a more accessible and refined level is desired. The certificate of the revision is attached.

Comments 4:quality of figures must be improved
Response 4:I sincerely apologize for not including the highest resolution images in the manuscript, primarily due to considerations regarding the overall size of the document. All original images possess a resolution exceeding 300 PPI and have been provided as supplementary attachments for your review. Following the feedback from subsequent experts concerning the font size and the emphasis of key elements within the images, I have made revisions to enhance their clarity and visibility.

Comments 5:Lines 215-217, I guess they are an oversight
Response 5:I express my sincere gratitude to the reviewer for their astute and meticulous examination of the manuscript. The content in question was originally part of the journal's template, and I had inadvertently overlooked its removal. I appreciate your reminder, and I have ensured its deletion in the revised version of the article.

Reviewer 2 Report

Comments and Suggestions for Authors

 In my opinion, the content of the manuscript is good enough. From a statistical point of view, there is nothing new in the work. Paper performed statistical analysis using standard methods. It is commendable that the authors even use the Kaplan-Meier method to study population survival.  I would just have a few comments about the presentation of the statistical results.
 (1) Figure 1. The text in the figure elements is actually invisible.
 (2) Table 1, Table 2. Spaces between table columns should be reduced.
 (3) Figure 2. Caplan-Meier curves are the most important characteristics
 in studying the survival of any population. However, the drawings
 provided are very small. They are virtually invisible.
 (4) Reference list. All entries should be presented in journal style

Author Response

Comments 1: Figure 1. The text in the figure elements is actually invisible.
Response 1:I concur with the reviewers' suggestion that the textual components in Figure 1 may have hindered the clarity of the image due to issues related to contrast and line thickness. We have undertaken a meticulous revision of the figure to enhance its clarity and comprehensibility.

Comments 2:Table 1, Table 2. Spaces between table columns should be reduced.
Response 2:I express my gratitude to the reviewers for their insightful suggestions. I have minimized the spacing between the two tables, thereby enhancing their overall compactness.

Comments 3:Figure 2. Caplan-Meier curves are the most important characteristics in studying the survival of any population. However, the drawings provided are very small. They are virtually invisible.
Response 3:The point raised by the reviewer is of paramount importance. The textual annotations and numerical data in the images I provided were rendered nearly illegible due to their diminutive size, thereby detrimentally affecting the viewers' experience. Consequently, I have revised the images to enhance clarity. The original images are submitted as attachments for reference.

Comments 4:Reference list. All entries should be presented in journal style
Response 4:I extend my sincere gratitude to the reviewer for their meticulous and thorough reading. In accordance with the journal's stylistic conventions, I have overlooked the stipulation regarding the punctuation following authors' names. I have ensured that all authors' names are now concluded with a semicolon. Furthermore, I have rectified the formatting of all references (Line 420-500).

Reviewer 3 Report

Comments and Suggestions for Authors

Comment 1:There are instances of improper or omitted use of articles (a/an/he) in multiple places.

Comment 2:The introduction should more clearly define the scientific gaps that this study aims to fill.

Comment 3:Curve graphs (such as Figure 2) and other charts can be improved through clearer annotations and labels.

Comment 4:The innovative description of one's own research in the paper is relatively vague. The specific innovative points of the study in terms of methods, theoretical models, or clinical applications were not explained in detail.

Comment 5:Hyperspectral systems also play an important role in disease diagnosis. For example, A stare-down video-rate high-throughput hyperspectral imaging system and its applications in biological sample sensing, it is suggested that hyperspectral systems should be discussed.

Comment 6:The author should carefully check their English. When describing the design and results of research, verb tenses can sometimes be inconsistent. Additionally, grammar and syntax errors throughout the manuscript should be addressed to enhance clarity and readability. For example, consistent use of past tense when referring to completed experiments and results would improve the flow of the text.

Author Response

Comment 1:There are instances of improper or omitted use of articles (a/an/he) in multiple places.
Response 1:We express our sincere gratitude to the reviewers for their meticulous and comprehensive examination of our manuscript. We have rectified the inappropriate usages within the text, highlighting these amendments in red for clarity. Furthermore, we have sought the assistance of MDPI for further linguistic refinement (english-84473).

Comment 2:The introduction should more clearly define the scientific gaps that this study aims to fill.
Response 2:This suggestion significantly contributes to enhancing the quality of our manuscript. In our introduction, we have incorporated the following passage to clearly articulate the gap in literature regarding the absence of reported effective combined therapeutic regimens for patients with HCC accompanied by PVTT. The revised text is as follows:
TACE in conjunction with targeted therapy represents a formidable strategy for the management of advanced HCC. Evidence suggests that this combined approach markedly improves OS when compared to either TACE or targeted therapy administered alone. Our previous research has also demonstrated that the integration of TACE with systemic therapy further enhances the overall response rate (ORR) and survival outcomes in patients with unresectable HCC. Nevertheless, the current evidence concerning the superiority of combination therapy over targeted therapy alone in HCC patients with PVTT remains scarce(Line 57-64).

Comment 3:Curve graphs (such as Figure 2) and other charts can be improved through clearer annotations and labels.
Response 3:The textual components of the figures may present challenges for reviewers in discerning the content due to issues related to contrast and line thickness. We have undertaken a meticulous revision of the images to enhance clarity and comprehensibility. The original high-definition images have been uploaded in the attachment section.

Comment 4:The innovative description of one's own research in the paper is relatively vague. The specific innovative points of the study in terms of methods, theoretical models, or clinical applications were not explained in detail.
Response 4:Previous studies have explored the efficacy and safety of localized therapies and combination regimens in the treatment of advanced hepatocellular carcinoma (HCC) (PMID: 37368105 and PMID: 38745965). However, our research delineates several distinguishing characteristics. Prior investigations often encompassed a heterogeneous population of advanced HCC patients, including those with portal vein invasion, lymph node metastasis, and various distant metastases. In contrast, our study exclusively focuses on cases of HCC associated solely with portal vein tumor thrombus (PVTT), systematically excluding patients with distant metastases or lymph node involvement, thereby targeting a more homogenous group. Moreover, we have conducted a profound subgroup analysis regarding the vascular supply of PVTT. This represents the inaugural statistical examination of targeted therapies or combination treatments predicated on the vascular characteristics of PVTT, enabling a stratified approach within the PVTT patient cohort. Our findings aim to identify subpopulations that may benefit most, thereby providing more precise clinical guidance. These additions were incorporated within line 306-318.

Comment 5:Hyperspectral systems also play an important role in disease diagnosis. For example, “A stare-down video-rate high-throughput hyperspectral imaging system and its applications in biological sample sensing”, it is suggested that hyperspectral systems should be discussed.
Comment 5:Hyperspectral systems also play an important role in disease diagnosis. For example, “A stare-down video-rate high-throughput hyperspectral imaging system and its applications in biological sample sensing”, it is suggested that hyperspectral systems should be discussed.
I am grateful to the reviewers for their valuable suggestions. Hyperspectral imaging systems play a significant role in disease diagnosis. Due to the inherent biological characteristics of hepatocellular carcinoma, most patients remain asymptomatic in the early stages; by the time symptoms manifest, the disease is often at an advanced stage, resulting in a negligible proportion of patients being suitable for curative treatments, alongside a dismal prognosis. Hyperspectral systems have demonstrated the capability to facilitate early, non-invasive diagnosis of liver cancer (PMID: 33376759), thereby accelerating the diagnostic process of tumor identification and enhancing classification precision (PMID: 31403260 + PMID: 36826944). Moreover, recent advancements in high-speed, wide-field infrared hyperspectral imaging (PMID: 38418468) have introduced further technological breakthroughs, suggesting a promising future role in the diagnosis of liver cancer. This manuscript focuses on patients with advanced liver cancer, whereas hyperspectral imaging systems are predominantly utilized in the context of early-stage liver cancer. The potential application of such systems in patient screening could facilitate the early detection of liver cancer, thereby preventing progression to advanced stages. This represents a promising avenue for future research and development.

Comment 6:The author should carefully check their English. When describing the design and results of research, verb tenses can sometimes be inconsistent. Additionally, grammar and syntax errors throughout the manuscript should be addressed to enhance clarity and readability. For example, consistent use of past tense when referring to completed experiments and results would improve the flow of the text.
Comment 6:We extend our gratitude to the reviewers for their meticulous and thorough evaluation. We have addressed the instances of improper usage within the manuscript, such as verb tense discrepancies, which have been rectified and highlighted in red. Additionally, we have sought further refinement through professional editing by MDPI (renglish-84473).

Reviewer 4 Report

Comments and Suggestions for Authors

The topic is of interest and the manuscript was well written. My comments:

1) The retrospective design represents a major limitation to the study. The authors could try to at least partially obviate to the risk of selection bias applying a propensity score matching.

2) The use of TACE in patients with PVT represents always a challenge due to the risk of ischemic injury to the liver. The authors should comment on that.

3) The authors should comment on the other systemic potential options in these patients (cite the recent SRMA: PMID: 34017396)

Author Response

Comments 1:The retrospective design represents a major limitation to the study. The authors could try to at least partially obviate to the risk of selection bias applying a propensity score matching.
Response 1:The reviewer has highlighted certain limitations in the statistical methodology employed in this study, which has greatly contributed to the enhancement of the manuscript. Given that this is a retrospective investigation, there may be inherent selection bias and potential discrepancies at baseline. However, our data indicate that the baseline characteristics between the two groups are well-balanced, with no statistically significant differences observed; thus, propensity score matching was deemed unnecessary. Nonetheless, I did attempt to conduct propensity score matching, utilizing a 1:2 ratio, which yielded analogous conclusions (please refer to the Non-published Material data before and after matching.docx).
Figure 1 illustrates the comparison of standardized mean differences (SMD) before and after matching, with a 1:2 matching ratio. Table 1 presents the baseline characteristics prior to and post-matching. Figure 2 and Table 2 provides the outcomes regarding overall survival (OS), progression-free survival (PFS), and objective response rate (ORR) subsequent to matching. The post-matching data corroborate the findings obtained prior to matching.

Comments 2:The use of TACE in patients with PVT represents always a challenge due to the risk of ischemic injury to the liver. The authors should comment on that.
Response 2:The reviewer has raised a pivotal point. Our findings lend support to the notion that TACE does not exacerbate the risk of hepatic failure in patients with HCC accompanied by PVTT. Anatomically, the liver receives a predominant portion of its blood supply via the portal vein. The presence of portal vein thrombosis poses a risk for hepatic ischemia and subsequent liver failure following TACE, which entails embolization of the hepatic artery. According to Chinese clinical guidelines, portal vein tumor thrombosis is not an absolute contraindication for TACE. However, in cases where main trunk thrombus leads to severe hepatic dysfunction, TACE should be avoided. The patients included in our study exhibited satisfactory hepatic reserve; despite imaging evidence of thrombus, collateral circulation had compensated over an extended period, rendering their liver function suitable for the consideration of TACE. This passage will be incorporated into the discussion (line 355-365).

Comments 3:The authors should comment on the other systemic potential options in these patients (cite the recent SRMA: PMID: 34017396)
Response 3:This constitutes a significant addition that enriches the depth of the discussion within the article. The discussion is as follows:
This article represents a retrospective study in which the systemic treatment protocols for patients were not uniformly established. The selection of systemic treatment regimens merits further deliberation, particularly given the multitude of therapeutic options for advanced HCC. The chronological advancement from the initial monotherapy of sorafenib, through lenvatinib, to various combination therapies underscores the complexity of first-line treatment selections, a matter that is continuously debated in terms of optimizing patient outcomes. Lenvatinib, as a multi-targeted inhibitor, demonstrates the highest ORR among single-target kinase inhibitors; however, the translation of this elevated ORR into prolonged OS benefits remains inadequately explored. A meta-analysis (PMID: 34017396) indicates that, while lenvatinib may yield superior ORR and PFS, these advantages do not necessarily confer survival benefits. Possible explanations for this discrepancy may include the proportion of patients receiving subsequent second-line therapies, as well as the impact of hepatic function impairment and other confounding factors influencing OS outcomes. This segment will be integrated into the discussion (Line 366-379).

Round 2

Reviewer 1 Report

Comments and Suggestions for Authors

the Authors did a very good job in addressing my comments. thank you.

Reviewer 3 Report

Comments and Suggestions for Authors

The authors have responded to my concerns

Reviewer 4 Report

Comments and Suggestions for Authors

The revised manuscript is OK for me. Thank you!